# Sleep Quality in Greek Adolescent Swimmers

**DOI:** 10.3390/jfmk9020087

**Published:** 2024-05-17

**Authors:** Vasileios T. Stavrou, George D. Vavougios, Glykeria Tsirimona, Zoe Daniil, Konstantinos I. Gourgoulianis

**Affiliations:** 1Laboratory of Cardio-Pulmonary Testing, Department of Respiratory Medicine, Medical School, University of Thessaly, 41110 Larissa, Greece; glukatsirimona@gmail.com (G.T.); zdaniil@uth.gr (Z.D.); 2RespiHub, ONISLOS MSCA COFUND, Department of Neurology, Medical School, University of Cyprus, 2417 Nicosia, Cyprus; 3Department of Neurology, Medical School, University of Cyprus, 2417 Nicosia, Cyprus; dantevavougios@hotmail.com

**Keywords:** swimming, sleep quality, young athletes, chest circumference, respiratory muscles

## Abstract

The aim of our study was to investigate the relationship between sleep quality and functional indices, swimming distance and gender in adolescent competitive swimmers. Forty-eight adolescent swimmers (boys, n = 22, 15.7 ± 1.0 years and girls, n = 26, 15.1 ± 0.8 years) were included in our study. They were assessed for handgrip strength, respiratory muscle strength and pulmonary function, answered a Pittsburg Sleep Quality Index questionnaire (PSQI), and recorded their anthropometric and morphological characteristics and training load for the last four weeks. The results showed differences between swimming distance and chest circumference difference, between maximal inhalation and exhalation (Δchest) (*p* = 0.033), PSQI score (*p* < 0.001), and sleep quality domains for “cannot breathe comfortably” (*p* = 0.037) and “have pain” (*p* = 0.003). Binary logistic regression (chi-square = 37.457, *p* = 0.001) showed that the variables Δchest (*p* = 0.038, 95% CI: 1.05–6.07) and PSQI score (*p* = 0.048, 95% CI: 0.1–1.07) remained independent predictors of the swim distance groups. Girls had a lower percentage of predicted values for the maximal inspiratory pressure (*p* < 0.001), maximal expiratory pressure (*p* = 0.027), forced expiratory volume within the first second (*p* = 0.026), forced vital capacity (*p* = 0.008) and sleep quality domains for “cough or snore loudly” (*p* = 0.032) compared to boys. A regression analysis showed that the sleep quality score was explained by the six independent variables: respiratory muscle strength (t = 2.177, β = 0.164, *p* = 0.035), Δchest (t = −2.353, β = −0.17, *p* = 0.023), distance (t = −5.962, β = −0.475, *p* < 0.001), total body water (t = −7.466, β = −0.687, *p* < 0.001), lean body mass (t = −3.120, β = −0.434, *p* = 0.003), and handgrip (t = 7.752, β = 1.136, *p* < 0.001). Our findings demonstrate that sleep quality in adolescent swimmers is a multifactorial result of morphometric characteristics, strength and respiratory function.

## 1. Introduction

Swimming is a popular individual sport for all ages. Swimmers divide their time between training, school, homework, social media and recovery [1]. Athletes who participate in official competitions and swimming championships have very demanding schedules [2]. According to Surda et al. [3], early morning sessions appear to be a major disruptive element of weekly sleep patterns. These weekly schedules require well-developed physical, mental and physiological characteristics and quality recovery to reduce the risk of fatigue [4] and/or overtraining. In addition, adolescents who do not get enough sleep are at higher risk of many health problems and are likely to have concentration and behavioural problems that can lead to poor school grades [5] and swimming performance [6]. Poor sleep quality affects many physiological characteristics and increases tension and confusion [7]. According to Chen et al. [8], adults with poor sleep quality have an increased secretion of catabolic hormones (cortisol) and changes in the pattern of secretion of anabolic hormones (testosterone), which has a negative effect on their muscle strength. Conversely, adequate sleep quality can increase muscle mass and prevent muscle loss [9]. In elite athletes, sleep quality is influenced by the type and intensity of exercise, as well as psychobiological mechanisms and chronotype, with negative or positive effects on athletic performance [10].

Previous studies have looked at training adaptations and the self-assessment of fatigue by athletes in the last weeks before a championship as an important tool for coaches to improve training planning [11,12], whereas pre-taper sleep is characterised as poorer in overreached swimmers and could contribute to different responses to the same training load. [13]. However, the sleep quality of Greek adolescent swimmers has not been addressed in the literature, so to this end, we designed this study in order to investigate the relationship between sleep quality and functional indices in adolescent swimmers. In addition, we aimed to address the results of sleep quality and examine possible differences among swimming distance and gender.

## 2. Materials and Methods

### 2.1. Participants

Forty-eight adolescent swimmers participated in the study (Table 1), six weeks before the Greek National Swimming Championship (NSC) and were divided into groups [gender: boys versus girls and swimming distance: <200 m versus ≥200 m]. The inclusion criteria were age ≥14 to ≤17 years; daily training for the last 4 years and participation in ≥2 events of the NSC; no recent injury and/or pain; normal echocardiography (no myocardial hypertrophy); no respiratory diseases (e.g., asthma); with menarche in girls and Tanner stage score > 3; and no previous SARS-CoV-2 infection [14]. The study was conducted in accordance with the Helsinki Declaration for Human Subjects (University of Thessaly, Greece: approval ref: No. 58076/22 November 2018) and written informed consent was obtained from all participants.

### 2.2. Data Collection

Data were collected two hours before the start of training in a quiet room. Height was measured using Seca 700 (Hamburg, Germany) and we calculated the Δchest as the difference between chest circumference at maximum inhalation and exhalation using Seca 201 (Seca, Hamburg, Germany). Body mass, lean body mass and total body water were estimated via bioelectrical impedance analysis (Tanita MC-980, Arlington Heights, IL, USA) and body surface area (BSA) [15] and body mass index (BMI) were calculated.

The grip strength test was assessed using an electronic dynamometer (Camry, EH 101, South El Monte, CA, USA) from a seated position with the arm at the side and the elbow at 90°. Each athlete performed three maximal isometric efforts for 5 s with both hands, alternating and in random order, with 40 s rest in between, and the average of the best left-and-right hand effort was calculated. [16]. A portable device was used to assess respiratory muscle strength (RMS = MIP / MEP) [maximal inspiratory pressure (MIP) and maximal expiratory pressure (MEP), AirOFit PRO™, AirOFit, Copenhagen, Denmark). Each swimmer performed three maximal trials for MIP and MEP, respectively, with 40 s of rest in between, and we recorded the highest score from a seated position. The predicted values for MIP and MEP were calculated [17] as follows:MIP (cmH_2_O) = 142 − (1.03 × Age(yrs))
MEP (cmH_2_O) = 180 − (0.91 × Age(yrs))

Lung function was assessed using a portable device (Card Guard Spiro Pro, Israel) and three maximal flow-volume loops were performed in the sitting position for each athlete, and the best attempt was evaluated [forced expiratory volume in 1 s (FEV_1_), forced vital capacity (FVC) and peak expiratory flow (PEF) [18]. All participants completed a validated questionnaire, the Pittsburgh Sleep Quality Index (PSQI) and were provided clarification where necessary [19].

Training load for the last four weeks was recorded as a percentage of: (a) aerobic exercise at an intensity of 60–70% of heart rate maximum (HR_max_), (b) anaerobic exercise at an intensity of 70–90% of HR_max_, (c) strength exercise (e.g., paddle, board, pull buoy, resistance belt, etc.), (d) hypoxic exercise at an intensity >80% of HR_max_, and (e) technique and skill exercise at an intensity <60% of HR_max_ (Table 1). Information on training load was provided by the athletes’ coaches.

### 2.3. Statistical Analysis

For statistics, the Independent Samples *t*-Test was used to test the differences between groups of genders (boys versus girls) and swimming distances (<200 m versus ≥200 m), and Cohen’s d was calculated from the mean difference between groups (M1 and M2), using 115 and the pooled standard deviation (SD): 



$$\mathrm{Cohen’s}\;d\;=\;\frac{M2-M1}{{SD}_{pooled}}\;{\mathrm{SD}}_{\mathrm{pooled}}\;=\sqrt{\frac{{SD1}^{2}+{SD2}^{2}}{2}}$$



Relationships between continuous variables were assessed using Pearson’s R correlation coefficients. A Chi-Square Test, modified by Fisher’s Exact Test with cell frequencies less than 5, was used to assess associations between qualitative variables. Binary Logistic Regression for the multivariate analyses of independent associations between these groups. For all tests, a *p*-value of <0.05 was considered statistically significant. The IBM SPSS 21 statistical package (SPSS Inc., Chicago, IL, USA) was used for all statistical analyses.

## 3. Results

The results between groups are shown in Table 2. Statistically significant differences were found between swim distance (n = 24 < 200 m ≥ n = 24) and Δchest (t_(46)_ = −2.204, *p* = 0.033, Cohen’s d = 0.64, medium effect size), PSQI score (t_(46)_ = 5.164, *p* < 0.001, Cohen’s d = 1.51, large effect size) and sleep quality domains for the questions "cannot breathe comfortably" (t_(46)_ = 2.145, p = 0.037, Cohen’s d = 0.62, medium effect size) and ‘have pain’ (t_(46)_ = 3.122, *p* = 0.003, Cohen's d = 0.90, large effect size). A weak negative significant correlation was found between Δchest and PSQI score (r = −0.323, *p* = 0.025). RMS was found to have a moderate negative relationship between genders (r = −0.533, *p* < 0.001), with a moderate positive relationship for lean body mass (r = −0.404, *p* = 0.004), a weak positive relationship for total body water (r = −0.339, *p* = 0. 018) a weak positive relationship for sleep quality for the question “cannot breathe comfortably” (r = −0.361, *p* = 0.012), as well as a moderate positive relationship for ‘have to get up to use the bathroom’ (r = −0.675, *p* < 0.001) and a moderate positive relationship for ‘have bad dreams’ (r = −0.417, *p* = 0.003).

Significant differences between Δchest (n = 20 < 8 cm ≥ n = 28) were found between handgrip strength (31.3 ± 6.5 versus 38.5 ± 7.9 kg, t_(46)_ = 3.344, *p* = 0.002, Cohen’s d = 0.99, large effect size) and sleep quality domain for “waking up in the middle of the night or early in the morning” (1.1 ± 0.6 versus 0.6 ± 0.6 score, t_(46)_ = −3.000, *p* = 0.004, Cohen’s d = 0.89, large effect size), with boys having higher scores compared to girls. Binary logistic regression (χ^2^ = 37.457, *p* = 0.001) showed that the variables Δchest (*p* = 0.038, 95% CI: 1.05–6.07) and PSQI score (*p* = 0.048, 95% CI: 0.1–1.07) remained independent predictors of swimming distance groups.

A regression analysis was used to explore the interpretation of the PSQI score. The multiple R of the regression was 0.81, which is statistically significantly different from zero, F(6, 41) = 28.645, *p* < 0.001. The variability of the sleep quality score was explained by the six independent variables: respiratory muscle strength (t = 2.177, β = 0.164, *p* = 0.035), Δchest (t = −2.353, β = −0.17, *p* = 0.023), distance (t = −5.962, β = −0.475, *p* < 0.001), total body water (t = −7.466, β = −0.687, *p* < 0.001), lean body mass (t = −3.120, β = −0.434, *p* = 0.003), and handgrip (t = 7.752, β = 1.136, *p* < 0.001).

## 4. Discussion

The data from the present study show that the Δchest parameter and PSQI score are related to swimming distance >200 m, whereas athletes with greater pain or breathing difficulties compete at shorter distances, probably due to sleep duration and type of exercise, findings which are consistent with a previous study by Stavrou et al. [20] in fin swimmers.

Our results showed that Δchest was associated with handgrip strength and sleep domains, with boys having higher values compared to girls. According to Nedelec et al. [10,16], sleep quality in elite athletes is influenced by the type and intensity of exercise, psychobiological mechanisms, chronotype and circadian rhythm. Circadian rhythm, according to Weger et al. [21], affects the physiological and biochemical functions of body systems, neurogenesis and is related to homeostasis. Our results showed a relationship between sleep quality and total body water, most likely due to induced dehydration, either from ambient temperature or from sleep disturbance, as measured by “cannot breathe comfortably” and “cough or snore loudly”. Our results also showed an association between sleep quality and respiratory muscle strength and handgrip strength. These findings are probably due to the low transmission of nerve impulses, leading to a lower potential energy transfer to muscle fibres, as observed in children with low biological development due to fatigue from high intensity training [10]. Although athletes in our study are classified as having good sleep quality (PSQI: 2.9 ± 1.7 score) compared to previous studies classifying poor sleep quality (PSQI: >5 score) [16].

Differences in respiratory muscle strength, morphological characteristics and sleep domains were observed between the sex groups in our study. These findings are likely due to differences in muscle loading and training (e.g., swimming style, distance, etc.), such as breath-holding training to increase anaerobic metabolism during hypoxic exercise [22]. A previous study by Stavrou et al. [22] reported that breathing frequency during exercise activates the parasympathetic system, particularly the vagus nerve innervating the lungs, while the signals transmitted by the vagus nerve, when the athlete develops voluntary hypoventilation, cause desaturation and stimulate the central chemoreceptors of the retrotrapezoid nucleus in the brainstem. It is common for swimmers to train via breath-holding, and this takes up a large proportion of the training session (Table 1), so the effect on the breathing pattern is an improvement in endurance and speed [22]. Thus, the increased PaCO_2_ produced during exercise decreases pH and shifts the oxygen–hemoglobin dissociation curve to the right, and as a result also increases the frequency and range of respiration to return to normal through the process of hyperventilation, causing the activation of anaerobic metabolism and leading to an increased production of lactic acid. According to Woorons et al. [23], the mechanism of the neutralisation of lactic acid from NaHCO_3_ plasma leads to the production of lactate solution and the release of CO_2_, while this hypoxic–hypercapnic environment (e.g., ↓ oxygen saturation, ↓ pH, ↑ lactic acid, ↑ CO_2_ production, etc.) affects the quality of sleep [24].

On the other hand, the parameters of respiratory muscle strength, Δchest, swimming distance, total body water, lean body mass and handgrip strength are a multifactorial combination that affects sleep quality during the special preparation period and disrupts sleep, according to our study results. These results are probably due to the interrelationship between sleep and exercise, which differentially affects the circadian rhythm and heart rate variability during exercise due to sympathetic nervous system stimulation, which improves the efficiency of sleep quality rather than sleep duration [25]. A previous study reported that insufficient sleep among adolescents is influenced by the use of technology and social media before bed and is a multifactorial condition related to biological, psychological and socio-cultural influences [26]. In addition, poor sleep may be due to an interrelationship between inflammatory processes and microtrauma injury from training or overtraining [27], lead to psychosomatic pain and/or due to low energy potential on Na+ and K+ channels and on actin–myosin coupling with cell membrane repolarisation in the receptor for the regulatory function of the cell, due to low hydration and/or poor nutrition [28].

In the present study, the adolescent athletes completed the PSQI questionnaire six weeks before the NSC, during the specific preparatory training period. PSQI scores are lowest during the competition phase, while athletes appear to be sleepier during the day and/or have a greater need for sleep due to training load and intensity [20]. Our results showed that athletes were classified as good sleepers according to the overall PSQI score (2.9 ± 1.7 points), but sleep quality decreased during the competition period due to anxiety, a change in environment, and the intensity of competition and training [29]. In addition, during periods of high intensity training, athletes report difficulty sleeping, restlessness during sleep, and leg fatigue [30], results that are compatible with the findings of our study.

### 4.1. Limitations, Strength, and Context

The interpretation of our results should be in the context of the limitations of our studies. There was no previous assessment during the season for athletes. In addition, the participants were aged between 14 and 17 years, and the different stages of puberty could possibly affect their behavioural development, different sleep patterns, physical growth and biological maturation [31]. Traditional methods for multiple comparisons adjustments focus on correcting for modest numbers of comparisons, often in an analysis of variance. A different set of techniques have been developed for “large-scale multiple testing”, in which thousands or even greater numbers of tests are performed. We did not consider adjustments for multiple comparisons on the premise of the exploratory design of the study with regard to the research question, i.e., “what is the physiological sleep quality of adolescent swimmers?” Our results should be interpreted in this context. Any differences reported should be investigated by targeted experiments. Another important limitation is that the results that are statistically significant may not be translated into swimming performance. Our results should be interpreted within this context, and each reported difference should be explored by targeted experiments. Another important limitation is that while statistically significant, this may not translate meaningfully in swimming performance.

Our study was the first to assess the relationship between sleep quality and functional indices in adolescent swimmers six weeks before the NSC. An interpretation of sleep quality through several control variables was found, which may be beneficial to swimming performance.

### 4.2. Practical Recommendations

○If you have trouble sleeping the night before, inform your coach so they can adjust your training load.○If you are a long-distance swimmer, check your sleep quality often.○For better sleep, keep hydrated.○Improve respiratory muscle strength for better sleep.

## 5. Conclusions

In conclusion, the findings of our study showed that the Δchest circumference difference between maximal inhalation and exhalation (Δchest) and the sleep quality score are affected in short distance swimmers (≤200 m) compared to middle-to-long distance swimmers (>200 m). The quality of sleep is a multifactorial outcome of morphometric characteristics, strength and respiratory function. Assessing sleep quality has practical applications for advising athletes and coaches, so the information provided can help structure and load training.

## Figures and Tables

**Table 1 jfmk-09-00087-t001:** Athletes’ and training characteristics. Data are expressed as mean ± standard deviation, percentage and number (n).

		Total	Range of Values	
Variable	Unit	(n = 48)	Lower	Upper
Age	years	15.2 ± 0.9	14.0	17.0
Body mass index	kg/m^2^	20.6 ± 1.8	17.3	24.6
Body surface area	m^2^	1.4 ± 0.2	1.1	1.9
Lean body mass	%	66.0 ± 4.6	58.6	74.6
Total body water	%	58.4 ± 7.4	49.2	71.0
Δchest	cm	7.8 ± 2.7	3.0	13.0
Handgrip	kg^−1^	35.5 ± 8.1	22.2	55.7
SpO_2_ resting	%	98.7 ± 0.5	98.0	99.0
Heart rate resting	% of predicted	39.7 ± 5.8	33.0	50.0
MIP	% of predicted	109.1 ± 11.1	97.0	133.0
MEP	% of predicted	105.5 ± 9.6	98.0	142.0
FEV_1_	% of predicted	111.5 ± 9.9	100.0	136.0
FVC	% of predicted	113.0 ± 10.5	94.0	162.0
PEF	% of predicted	142.3 ± 27.5	99.0	156.0
PSQI	score	3.0 ± 1.7	0.0	7.0
Swimming style/distance	BK: 50 m (B, n = 4, G, n = 4), 100 m (B, n = 2), 200 m (B, n = 2); BR: 50 m (B, n = 2, G, n = 2), 100 m (G, n = 2), 200 m (B, n = 2, G, n = 2); BF: 100 m (G, n = 2), 200 m (B, n = 2, G, n = 2); FR: 50 m (G, n = 2), 100 m (B, n = 2, G, n = 2), 200 m (B, n = 2), 400 m (G, n = 2), 1500 m (B, n = 2, G, n = 2); IM: 400 m (B, n = 4, G, n = 2)			
Type of exercise/percent of each training	Aerobic = 15%; Anaerobic = 20%; Strength = 15%; Hypoxic = 35%; Technique and Skills = 15%			

Abbreviations: B = boys; BF = butterfly; BK = backstroke; BR = breaststroke; FEV_1_ = forced expiratory volume in 1st s; FR = freestyle; FVC = forced vital capacity; G = girls; IM = individual medley; MEP = maximal expiratory pressure; MIP = maximal inspiratory pressure; PEF = peak expiratory force; PSQI = Pittsburgh Sleep Quality Index; SpO_2_ = oxygen saturation with pulse oximetry; Δchest = chest circumference difference between maximal inhalation and exhalation.

**Table 2 jfmk-09-00087-t002:** Results between groups. Data are expressed as mean ± standard deviation and percentage.

Variable		Athletes	Gender		*p* Value	Swimming Distance		*p* Value
	Unit	Total	Boys (n = 22)	Girls (n = 26)		≤200 m (n = 24)	>200 m (n = 24)	
Age	years	15.2 ± 0.9	15.7 ± 1.0	15.1 ± 0.8	0.661	15.3 ± 1.1	15.2 ± 0.7	0.758
Body mass index	kg/m^2^	20.6 ± 1.8	20.2 ± 1.7	20.9 ± 1.9	0.152	21.0 ± 1.9	20.2 ± 1.7	0.162
Body surface area	m^2^	1.4 ± 0.2	1.5 ± 0.2	1.4 ± 0.2	0.003	1.4 ± 0.2	1.4 ± 0.2	0.584
Lean body mass	%	50.5 ± 3.3	68.6 ± 3.6	63.8 ± 4.2	<0.001	65.3 ± 4.3	66.6 ± 4.7	0.319
Total body water	%	58.3 ± 7.4	64.6 ± 4.9	53.0 ± 4.2	<0.001	58.7 ± 7.9	57.9 ± 6.9	0.741
Δchest	cm	7.8 ± 2.4	8.6 ± 2.7	7.0 ± 2.6	0.037	6.9 ± 2.4	8.5 ± 2.7	0.033
Handgrip	kg	35.5 ± 8.1	41.3 ± 7.1	30.6 ± 5.1	<0.001	36.0 ± 8.7	35.0 ± 7.6	0.678
MIP	% of predicted	109.1 ± 11.0	103.7 ± 11.9	98.6 ± 11.8	<0.001	97.3 ± 10.8	98.0 ± 9.8	0.803
MEP	% of predicted	105.3 ± 9.6	98.6 ± 11.8	92.9 ± 4.5	0.027	94.3 ± 5.3	96.8 ± 11.6	0.359
FEV_1_	% of predicted	111.6 ± 9.8	108.8 ± 12.7	102.7 ± 4.5	0.026	104.1 ± 4.5	106.9 ± 12.8	0.313
FVC	% of predicted	113.1 ± 20.5	115.5 ± 10.9	108.2 ± 10.7	0.008	112.0 ± 11.9	111.0 ± 7.5	0.751
PEF	% of predicted	142.4 ± 27.5	146.0 ± 27.3	139.2 ± 27.7	0.400	143.1 ± 24.0	141.6 ± 30.8	0.852
PSQI	score	2.9 ± 1.7	2.8 ± 1.8	3.1 ± 1.7	0.613	4.0 ± 1.8	1.9 ± 0.8	<0.001
cannot get to sleep within 30 min	score	0.9 ± 0.6	1.0 ± 0.6	0.8 ± 0.5	0.363	0.9 ± 0.6	0.9 ± 0.5	0.983
wake up in the middle of the night or early morning	score	0.7 ± 0.7	0.8 ± 0.6	0.7 ± 0.7	0.798	0.8 ± 0.7	0.7 ± 0.6	0.662
have to get up to use the bathroom	score	0.5 ± 0.8	0.7 ± 0.9	0.3 ± 0.4	0.060	0.6 ± 0.6	0.4 ± 0.8	0.460
cannot breathe comfortably	score	0.1 ± 0.3	0.1 ± 0.3	0.1 ± 0.3	0.865	02 ± 0.4	0.0 ± /	0.037
cough or snore loudly	score	0.1 ± 0.4	0.3 ± 0.6	0.0 ± /	0.032	02 ± 0.6	0.8 ± 0.3	0.521
feel too cold	score	0.1 ± 0.2	0.1 ± 0.3	0.0 ± /	0.121	0.0 ± /	0.1 ± 0.3	0.155
feel too hot	score	0.6 ± 0.9	0.7 ± 1.0	0.6 ± 0.7	0.675	1.1 ± 0.9	0.3 ± 0.6	0.001
have bad dreams	score	0.3 ± 0.6	0.4 ± 0.7	0.3 ± 0.6	0.763	0.4 ± 0.6	0.3 ± 0.6	0.365
have pain	score	0.2 ± 0.5	0.2 ± 0.4	0.2 ± 0.6	0.733	0.4 ± 0.7	0.0 ± /	0.003

Abbreviations: FEV_1_ = forced expiratory volume in 1st s; FVC = forced vital capacity; MEP = maximal expiratory pressure; MIP = maximal inspiratory pressure; PEF = peak expiratory force; Δchest = chest circumference difference between maximal inhalation and exhalation.

## Data Availability

Please contact author for data requests.

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
