# Peer review of "Sleep Quality in Greek Adolescent Swimmers"

_jfmk, 2024, doi:10.3390/jfmk9020087_

Round 1
Reviewer 1 Report
Comments and Suggestions for Authors
The field study aimed to investigate the interrelationship between the quality of sleep and morpho-functional indices, swimming distance, and gender, in adolescent high-level swimmers. Forty-eight adolescent swimmers, having > 4 years of training, ≥ 60 min of training hours per week for the last two years, among them 22 boys in the age 15.7±1.0 years, and 26 girls in the age 15.1±0.8 years were included in our study. The handgrip strength, respiratory muscle strength, pulmonary function test, quality of sleep (PSQI), anthropometric and morphological characteristics, and training load for the last four weeks were assessed. Results showed differences between swimming distance and Δchest (p=0.033), PSQI score (p<0.001), and domains of quality of sleep for the questions “cannot breathe comfortably” (p=0.037) and “have pain” (p=0.003). Binary logistic regression (χ2 =37.457, p=0.001) indicated that the variables Δchest (p=0.038, 95 % CI: 1.05-6.07) and PSQI score (p=0.048, 95 % CI: 0.1-1.07) remained independent predictors of swimming distance groups. Regression analysis showed that of sleep quality score was explained by the six independent variables: respiratory muscle strength, Δchest, distance, total body water, lean body mass, and handgrip.
Finally, the authors concluded that the chest circumference difference between maximal inhalation and exhalation (Δchest) and sleep quality score in short swimming distances (≤200m) athletes compared to athletes swimming middle-to-long distances (>200m) were affected. The sleep quality occurred to be a multifactorial result of morphometric characteristics, strength, and respiratory function. Study limitations: lack of previous assessment during the season in athletes, and participants aged between 14 and 17 are possibly the different stages of puberty.
Reviewer`s opinion: the title seems to be too long, and the last part of the title - the sentence „Field study for health and performance” should be removed. We don`t know which were differences between boys and girls in all parameters – this should be compared and statistically calculated. There is no conclusion in the abstract - this must be inserted. The manuscript is written in poor English – and should be improved.
Author Response
We thank the reviewer for the comments that have helped us to improve the paper. All changes have been indicated by red color within the text. Below you will find a point-by-point response to your comments.
Comments 1. the title seems to be too long, and the last part of the title - the sentence „Field study for health and performance” should be removed.
Response: We thank the reviewer for his/her suggestion. It was removed.
Comments 2. We don`t know which were differences between boys and girls in all parameters – this should be compared and statistically calculated.
Response: We thank the reviewer for his/her comment. The differences between boys and girls are shown in Table 2 and have been included in the abstract.
Comments 3. There is no conclusion in the abstract - this must be inserted.
Response: We thank the reviewer for his/her suggestion. It was added.
Comments 4. The manuscript is written in poor English – and should be improved.
Response: We thank the reviewer for his/her comment. We improved the English in manuscript.
Comments 5. Abstract - report all significant results with p values.
Response: We thank the reviewer for his/her suggestion. It was added.
Reviewer 2 Report
Comments and Suggestions for Authors
Dear All,
I hope my comments are helpful in improving your work.
Reading the introduction, I think there are many claims that need to be supported by authors. For example, between lines 34 and 37 there are no quotes that support the statements being made. Please read the entire introduction and add citations where appropriate.
Regarding the information that is provided, and taking into account the objective that they set, I believe that there is a lack of more information on all the variables that work, and that they also provide more background on the problem that exists, the greatest damage that this problem can bring. , that is, highlight the importance of why it is necessary to carry out your research.
As for the objective, I think this should be in the same line as its title. In the first he talks about inter-relation, and the second about the effect. Please determine what you did and modify where appropriate.
The title also talks about morpho-functional indices, but the objective is morphological and functional characteristics. Please correct where appropriate.
Regarding the inclusion criteria, what is the justification for those you specify? (I think it's because that indicates that they are high-performance athletes, but I think they should specify it).
I have a doubt about the indices, I think they are those specified in table 1, but it is not clear to me. I think this should be better specified. If I am correct, I think it is necessary to explain in methodology the way in which these indices are calculated.
Among the indices is body composition? I ask this because when reading point 2.2 I see that the TANITA instrument is used.
I think that for a better understanding, in the methodology chapter all the variables worked on should be specified in greater detail.
The title in bold is missing from the chapter RESULTS:
In general, the discussion chapter seems correct to me. I think that the newness that its study contributes should be highlighted to a greater extent, both to knowledge and to the practical part.
They should also add the strengths of their study, and new lines of research that could be worked on (these new lines should be related to the limitations that their study had, and the results found).
I think that the conclusions should be better worked, since they must respond to their objective, and since they have hypotheses, it should be said whether the hypotheses were accepted or rejected.
Greetings.
Author Response
We thank the reviewer for the comments that have helped us to improve the paper. All changes have been indicated by red color within the text. Below you will find a point-by-point response to your comments.
Comments 1. Reading the introduction, I think there are many claims that need to be supported by authors. For example, between lines 34 and 37 there are no quotes that support the statements being made. Please read the entire introduction and add citations where appropriate.
Response: We thank the reviewer for his/her comment. It was added references.
Comments 2. Regarding the information that is provided, and taking into account the objective that they set, I believe that there is a lack of more information on all the variables that work, and that they also provide more background on the problem that exists, the greatest damage that this problem can bring. , that is, highlight the importance of why it is necessary to carry out your research.
Response: We thank the reviewer for his/her suggestion. Information has been added at the end of the introduction.
Comments 3. As for the objective, I think this should be in the same line as its title. In the first he talks about inter-relation, and the second about the effect. Please determine what you did and modify where appropriate.
Response: We thank the reviewer for his/her comment. It was replaced.
Comments 4. The title also talks about morpho-functional indices, but the objective is morphological and functional characteristics. Please correct where appropriate.
Response: We thank the reviewer for his/her suggestion. It was corrected the title.
Comments 5. Regarding the inclusion criteria, what is the justification for those you specify? (I think it's because that indicates that they are high-performance athletes, but I think they should specify it).
Response: We thank the reviewer for his/her comment. It was rewritten.
Comments 6. I have a doubt about the indices, I think they are those specified in table 1, but it is not clear to me. I think this should be better specified. If I am correct, I think it is necessary to explain in methodology the way in which these indices are calculated.
Response: We thank the reviewer for his/her suggestion. The obscure parts of the methodology have been explained and rewritten.
Comments 7. Among the indices is body composition? I ask this because when reading point 2.2 I see that the TANITA instrument is used.
Response: We thank the reviewer for his/her comment. The TANITA was used to estimate body composition via bioelectrical impedance.
Comments 8. I think that for a better understanding, in the methodology chapter all the variables worked on should be specified in greater detail.
Response: We thank the reviewer for his/her suggestion. Where necessary, clarifications have been added to the methodology.
Comments 9. The title in bold is missing from the chapter RESULTS:
Response: We thank the reviewer for his/her comments. It was corrected the typo.
Comments 10. In general, the discussion chapter seems correct to me. I think that the newness that its study contributes should be highlighted to a greater extent, both to knowledge and to the practical part.
Response: We thank the reviewer for his/her suggestion. It was added “Recommendations” section at the end of discussion.
Comments 11. They should also add the strengths of their study, and new lines of research that could be worked on (these new lines should be related to the limitations that their study had, and the results found).
Response: We thank the reviewer for his/her suggestion. It was added “Strength and limitations” section at the end of discussion.
Comments 12. I think that the conclusions should be better worked, since they must respond to their objective, and since they have hypotheses, it should be said whether the hypotheses were accepted or rejected.
Response: We thank the reviewer for his/her comment. It was corrected.
Round 2
Reviewer 2 Report
Comments and Suggestions for Authors
Dear friends, your work has improved and I believe it is ready to be published.
Congratulations.
Author Response
Thank you